# MCAs in Arabidopsis are Ca$^{2+}$-permeable mechanosensitive channels inherently sensitive to membrane tension

Kenjiro Yoshimura [1✉], Kazuko Iida[2,3] & Hidetoshi Iida [2✉]

Mechanosensitive (MS) ion channels respond to mechanical stress and convert it into intracellular electric and ionic signals. Five MS channel families have been identified in plants, including the Mid1-Complementing Activity (MCA) channel; however, its activation mechanisms have not been elucidated in detail. We herein demonstrate that the MCA2 channel is a Ca$^{2+}$-permeable MS channel that is directly activated by membrane tension. The N-terminal 173 residues of MCA1 and MCA2 were synthesized in vitro, purified, and reconstituted into artificial liposomal membranes. Liposomes reconstituted with MCA1(1-173) or MCA2(1-173) mediate Ca$^{2+}$ influx and the application of pressure to the membrane reconstituted with MCA2(1-173) elicits channel currents. This channel is also activated by voltage. Blockers for MS channels inhibit activation by stretch, but not by voltage. Since MCA proteins are found exclusively in plants, these results suggest that MCA represent plant-specific MS channels that open directly with membrane tension.

[1] Department of Machinery and Control Systems, College of Systems Engineering and Science, Shibaura Institute of Technology, Saitama 337-8570, Japan. [2] Department of Biology, Tokyo Gakugei University, 4-1-1 Nukuikita-machi, Koganei, Tokyo 184-8501, Japan. [3] Laboratory of Biomembrane, Tokyo Metropolitan Institute of Medical Science, 2-1-6 Kamikitazawa, Setagaya, Tokyo 156-8506, Japan. ✉email: kenjiroy@shibaura-it.ac.jp; iida@u-gakugei.ac.jp

Mechanosensitive (MS) channels are integral membrane proteins with gated pores that mediate the flux of ions across membranes in response to extrinsic mechanical stress, such as touch, wind, gravity, osmotic pressure, and pathogen invasion, as well as intrinsic mechanical stress produced by cell division or cellular growth[1,2]. Despite their importance in responses to mechanical stress, the precise mechanisms for the gating of MS channels remain unclear. However, the following two general mechanisms have been proposed and are widely accepted: the force-from-lipids mechanism[3,4], in which MS channels act as direct mechanosensors of tension in the lipid bilayer, and the force-from-filaments mechanism[5,6], in which the extracellular matrix and/or cytoskeleton act as mediators of the force that pulls MS channels open.

Five families of MS channels have been reported to date in plants: Mid1-Complementing Activity (MCA) proteins[7,8], MscS-like (MSL) proteins[9,10], Two-Pore Domain Potassium (TPK) channels[11], Reduced hyperosmolality-induced $[Ca^{2+}]_i$ increase (OSCA) channels[12] evolutionarily related to DUF221[13,14], and Piezo channels[15]. MCA channels are structurally and evolutionarily unique. Arabidopsis (Arabidopsis thaliana) MCA proteins have a single transmembrane segment and assemble into a homotetramer to constitute a channel[16–18], while the subunits of other plant MS channel families have multiple transmembrane segments and form multimers, including dimers (TPK), trimers (Piezo), pentamers (OSCA), and heptamers (MSL)[19–21]. Genes encoding MCA proteins are found exclusively in plants[7,22,23], while those of other plant MS channel homologs are present in archaea, bacteria, protists, fungi (MSL), and animals (Piezo, OSCA, and TPK)[9,11,12,15,24,25]. MCA1 and MCA2 are involved in the response to gravity[26,27] and cold[28] in Arabidopsis. Moreover, MCA1 is required to perceive the hardness of soil and hypoosmotic stress[7]. Of the entire MCA1 and MCA2 protein chains comprising 421 and 416 residues, respectively (Fig. 1a), the N-terminal 173 residues, including the transmembrane segment and EF-hand-like motif, are sufficient for $Ca^{2+}$ uptake activity when expressed in yeast cells[17]. Therefore, a $Ca^{2+}$-permeable pore is likely to be composed of these 173 residues.

In Plantae, Arabidopsis MCA1 and MCA2 have distinct and overlapping expression patterns[29]. Histochemical assays for the β-glucuronidase reporter fused to the MCA1 or MCA2 promoter have shown that MCA1 and MCA2 are expressed in the vascular tissues of the primary root, cotyledons, and leaves and in the stele and endodermis, but not in the cortex or epidermis of the primary root. Furthermore, both proteins are expressed in a region corresponding to the shoot apical meristem, but not in the root cap or hair. Differential expression patterns have also been observed: MCA1 is expressed in the promeristem and adjacent elongation zone of the primary root, whereas MCA2 is not. In contrast, MCA2 is expressed in the mesophyll cells of cotyledons and leaves, whereas MCA1 is not. The subcellular localization of green fluorescent protein fused C-terminally to MCA1 or MCA2 revealed that both proteins are present in the plasma membrane. In addition, there is no data about potential heteromerization of MCA1 and MCA2[17]. These distinct expression patterns suggest that MCA1 and MCA2 play different physiological roles in plants.

The demonstration of direct activation by membrane tension requires the functional reconstitution of purified channels into an artificial lipid bilayer and activation by membrane stretch. Arabidopsis OSCA1.2 was found to be inherently MS in this type of experiment[30]. Although MSL, Piezo, and TKP in plants have not been demonstrated to be inherently MS, their counterparts in bacteria and animals have been[31–33]. The present study was performed to investigate whether MCA, which is unique to plants, has the gating capacity to be directly activated by membrane tension. We synthesized MCA1 and MCA2 in vitro, reconstituted them into liposomes, and examined $Ca^{2+}$ permeability and electrophysiological properties. We herein report a C-terminally truncated form of the MCA2 protein that acts as a $Ca^{2+}$-permeable, inherently mechanosensitive channel, which directly senses membrane tension for channel opening.

## Results

**In vitro synthesis of MCA1 and MCA2 proteins.** To obtain purified MCA proteins, we used a cell-free expression system for recombinant proteins instead of cell-based synthesis systems. This is because full-length and truncated MCA proteins are harmful to host cells when expressed in large amounts[17,18]. This property seems to make it difficult to purify those proteins adequately for the analyses in the present study. A cell-free system also allows easy protein purifaction.

The MCA1 and MCA2 proteins were synthesized using a liposome-supplemented wheat germ cell-free translation system[34], in which synthesized membrane proteins are inserted into the lipid bilayers of liposomes. We initially attempted to synthesize and purify full-length MCA1 and MCA2 proteins fused C-terminally with a 6×His tag, designated MCA1-6H and MCA2-6H, respectively; however, since they underwent extensive aggregation, we were unable to solubilize them during subsequent purification steps. Therefore, we synthesized the N-terminal 173 residues with a His tag fused to their C termini, MCA1(1-173)-6H and MCA2(1-173)-6H, and found that both proteins aggregated to a markedly lesser extent. The synthesized proteins were solubilized with the non-ionic, amphiphilic detergent n-dodecyl-β-D-maltopyranoside (DDM; 1%) (Fig. 1b, lane 2) and applied to a nickel-nitrilotriacetic acid (Ni-NTA) column for one-step affinity purification. The eluate (Fig. 1b, lane 4) was then subjected to PD-10 desalting column chromatography to change the elution solution to a DDM-containing buffer suitable for the reconstitution of channels on liposomes. The purified MCA2(1-173)-6H proteins in the final preparation migrated on SDS-PAGE gels with a mobility corresponding to a molecular mass of approximately 19 kDa (Fig. 1b, lane 5), which is close to the protein mass (21.4 kDa) calculated from its amino acid sequence.

Using the same procedures, we also synthesized and purified the MCA1(1-173)-6H protein and two mutant proteins with a single amino acid substitution, MCA1(1-173)D21N-6H and MCA2(1-173)D21N-6H, the latter two of which were constructed by the site-directed mutagenesis of Asp-21 present in the unique transmembrane segment of MCA1(1-173) and MCA2(1-173) (Fig. 1a)[17,18]. Asp-21 has a critical carboxylic residue in the pore-lining helices and, thus, the D21N mutation results in the loss of $Ca^{2+}$ influx activity of full-length MCA1 and MCA2 in yeast cells[17].

Figure 1c shows that the final preparations had no visible protein contaminants, except for a small amount of a protein with a slightly slower mobility, which always appeared with the same abundance ratio to the main protein during the purification steps, suggesting that the slower mobility protein is a consequence of its anomalous SDS-PAGE behavior often caused by altered detergent binding to membrane proteins[35]. Figure 1c also shows that MCA1(1-173)-6H and MCA1(1-173)D21N-6H migrated slower than MCA2(1-173)-6H and MCA2(1-173)D21N-6H in SDS-PAGE gels. MCA1(1-173) and its mutant protein appeared as 20-kDa bands, which is close to the protein mass (21.7 kDa) calculated from their amino acid sequences. To avoid lengthy names for the constructs, the abbreviation of 6 × His, 6H, was not added to the main text hereafter.

**C-terminally truncated MCA1 and MCA2 form tetramers.** Previous studies demonstrated that full-length MCA1 and MCA2

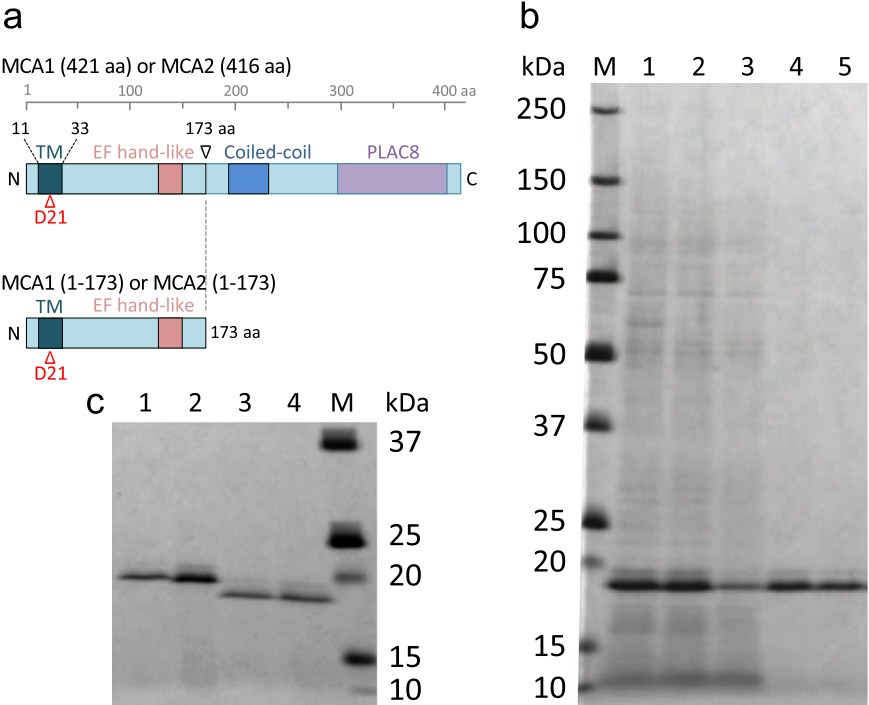

**Fig. 1 Schematic diagram and purification of MCA proteins. a** Schematic diagram of intact MCA1 and MCA2 and their C-terminally truncated proteins. TM, transmembrane segment. D21, Asp-21 crucial for $Ca^{2+}$ influx[17]. **b** Purification of MCA2(1-173)-6H. Protein samples were subjected to SDS-PAGE followed by silver staining. Lane 1, A crude proteoliposome fraction. Lane 2, 1% DDM-solubilized sample. Lane 3, A flow-through fraction from a Ni-NTA column. Lane 4, An eluate from the same column eluted with 300 mM imidazole. Lane 5, An eluate from a PD-10 desalting column eluted with MOPS buffer described in the Methods section. One-4000th of each sample was loaded onto the respective lanes. M, Molecular weight markers. A typical result from six independent experiments is shown. **c** Purified MCA proteins used for reconstitution into liposomes. Each sample is an eluate from the PD-10 desalting column described above. Lane 1, MCA1(1-173)D21N-6H; Lane 2, MCA1(1-173)-6H; Lane 3, MCA2(1-173)D21N-6H; Lane 4, MCA2(1-173)-6H. A typical result from three independent experiments is shown. Note that the molecular weights of MCA1(1-173)-6H and MCA2(1-173)-6H were calculated from their amino acid sequences to be 21.7 and 21.4 kDa, respectively. For comparison, note that the molecular weights of MCA1(1-173) and MCA2(1-173) tagged with four tandem hemagglutinin repeats, which were previously used[17], but not in the present study, were calculated to be 27.8 and 27.5 kDa, respectively.

expressed in yeast and Sf9 insect cells form tetramers[17,18]. To investigate whether C-terminally truncated MCA1(1-173) and MCA2(1-173) proteins synthesized with a cell-free expression system form tetramers, we cross-linked the purified proteins with disuccinimidyl suberate (DSS) in the presence of DDM according to a method described elsewhere[36]. We then analyzed the reaction products using SDS-PAGE followed by silver staining (Fig. 2). In the absence of DSS, both proteins were detected as monomers, while its addition mainly generated three bands corresponding to the dimeric, trimeric, and tetrameric forms of the proteins (Fig. 2a, c). Importantly, a greater amount of the tetramer was observed at an increasing concentration of DSS. These results suggest that both proteins were mainly maintained in their tetrameric forms in DDM micelles. Equivalent results were obtained for the MCA1(1-173)D21N and MCA2(1-173)D21N mutant proteins (Fig. 2b, d), suggesting that the D21N mutation does not affect the tetrameric structure and also that MCA1(1-173)D21N and MCA2(1-173)D21N assemble properly, similar to wild-type MCA1(1-173) and MCA2(1-173). The band at approximately 250 kDa may be a cross-linked product of the aggregates mentioned above.

**C-terminally truncated MCA1 and MCA2 are permeable to $Ca^{2+}$.** Previous studies indicated that MCA1(1-173) and MCA2(1-173), which were tagged with four tandem hemagglutinin antigen repeats, were permeable to $Ca^{2+}$ when expressed in

yeast cells[17]. Therefore, we investigated whether the two proteins synthesized in vitro permeated $Ca^{2+}$ when incorporated into liposomes (hereafter referred to as MCA1(1-173) and MCA2(1-173) liposomes). The control group included liposomes prepared with cell-free protein synthesis reaction products from the empty vector (vector liposomes) and liposomes incubated without reaction products (pure liposomes). Liposomes were loaded with the fluorescent $Ca^{2+}$ indicator, Fluo-4, and fluorescence was monitored when $CaCl_2$ was added to the bath solution.

The bath solution initially contained 1 mM EGTA, which chelates $Ca^{2+}$. When the concentration of $CaCl_2$ in the bath solution was 0.5 mM, a small increase in fluorescence was detected in the suspensions of MCA1(1-173) and MCA2(1-173) liposomes (Fig. 3a). A $CaCl_2$ concentration of 1.5 mM induced a larger increase in fluorescence, while the further increase in fluorescence caused by 2.5 mM was small. In contrast, only a small and gradual increase in fluorescence was observed in vector and pure liposomes. The significantly larger increases observed in fluorescence in MCA1(1-173) and MCA2(1-173) liposomes than in control liposomes indicated an elevated $Ca^{2+}$ concentration due to its influx through MCA1(1-173) and MCA2(1-173). The largest increase between 0.5 and 1.5 mM was consistent with the 1:1 stoichiometry of the binding of $Ca^{2+}$ to EGTA.

As described above, the D21N mutation in MCA1 and MCA2 resulted in the loss of $Ca^{2+}$ uptake when expressed in yeast cells[17]. To clarify whether this loss found in vivo correlates with $Ca^{2+}$ influx into liposomes, we synthesized and purified MCA1(1-173)

D21N and MCA2(1-173)D21N (Fig. 1c). When both proteins were reconstituted into liposomes, the changes observed in fluorescence upon increases in the bath $Ca^{2+}$ concentration were similar to those in pure liposomes (Fig. 3b), suggesting that the mutant proteins do not mediate $Ca^{2+}$ influx into liposomes. The failure to detect the activities of the MCA1(1-173)D21N and MCA2(1-173) D21N proteins was unlikely to be due to these proteins not being incorporated into liposomes because MCA1(1-173)D21N and MCA2(1-173)D21N both formed tetramers in DDM micelles (Fig. 2). In addition, since Asn is less polar than Asp, the D21N mutation did not appear to interfere with the incorporation of the MCA(1-173) or MCA2(1-173) protein into liposomes. Collectively,

these results suggest that MCA1(1-173) and MCA2(1-173) have the ability to mediate $Ca^{2+}$ influx.

## Voltage-activation of C-terminally truncated MCA2.

The channel activity of MCA2(1-173) was examined using a patch-clamp technique. A large blister of a lipid bilayer was generated from liposomes reconstituted with MCA2(1-173). When a pipette potential of +160 mV was applied to the membrane excised from the blister, sparse current steps with a fixed amplitude were detected (Fig. 4a). When the holding potential was increased to +180 mV or higher, increases were observed in the frequency of the current and unit amplitude. Flickering was detected in the current and a stable subconducting state was not observed (Fig. 4b). In support of the absence of a stable subconducting state, the all-point histogram of the open state was fit with a single gaussian distribution (Supplementary Fig. 1). This current was observed in 59 out of 124 patches, but not (0/8 patches) when liposomes were prepared from the reaction product from an empty vector. The amplitude of currents increased by approximately two-fold when two steps appeared to overlap (see Fig. 5a).

When the holding potential was changed between −200 and +200 mV, channel opening was only detected at potentials larger than +100 mV and the open probability increased at higher voltages (Fig. 4c). Unit current increased almost linearly between +140 and +200 mV and the slope appeared to be shallower at lower potentials (Fig. 4d). Slope conductance changes with voltage have been reported for other channels[37]. When 200 mM KCl in the bath solution was replaced with 200 mM potassium gluconate for the assessment of ion selectivity, a reduction in conductance of approximately 30% was observed, suggesting the conductance of chloride ions at least in part (Fig. 4d).

Ion selectivity may be analyzed more quantitatively by assessing the reversal potential, namely, the membrane potential at which the direction of current reverses. However, reversal in the direction of the channel current was not observed because current occurred in one direction only and the current-to-voltage curve was unable to be reliably extrapolated to zero voltage (Fig. 4d). Although we attempted to preactivate the channel at a positive potential and then changed the holding potential to a negative potential using a protocol for recording the so-called tail current, channel opening was not detected after the potential was alternated to a negative potential (Fig. 4e).

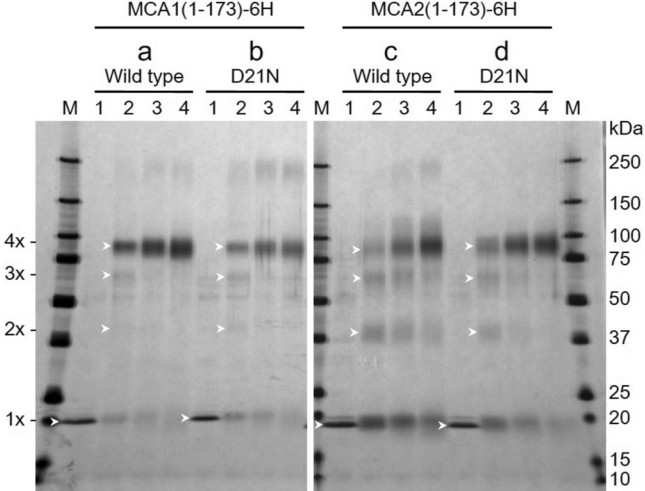

**Fig. 2 Tetramer formation by MCA1(1-173), MCA2(1-173), and their D21N mutant proteins synthesized with a cell-free translation system.** In vitro-synthesized wild-type and D21N proteins, which were similar to the protein preparation shown in Fig. 1c, were cross-linked for 1 h in the presence of DMM at room temperature with DSS at concentrations of 0 (lane 1), 0.1 (lane 2), 0.3 (lane 3), and 1.0 mM (lane 4). Reaction mixtures were subjected to SDS-PAGE followed by silver staining. **a** Wild-type MCA1(1-173)-6H, **b** MCA1(1-173)D21N-6H, **c** Wild-type MCA2(1-173)-6H, and **d** MCA2(1-173)D21N-6H. Arrowheads show a monomer, dimer, trimer, or tetramer. A typical result from two (left panel) or four (right panel) independent experiments is shown.

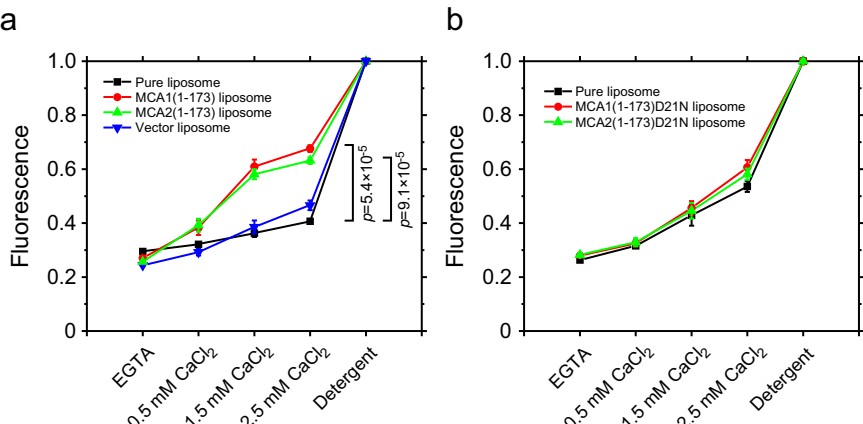

**Fig. 3 $Ca^{2+}$ permeation through MCA1(1-173) and MCA2(1-173) reconstituted into liposomes containing Fluo-4.** **a** Fluorescence from liposomes reconstituted with MCA1(1-173)-6H or MCA2(1-173)-6H or from control liposomes. Liposomes were preloaded with Fluo-4. $CaCl_2$ was sequentially added to the bath solution containing 1 mM EGTA. A detergent was added at the end of the experiment and fluorescence at this point was used to normalize fluorescence. **b** Fluorescence from liposomes reconstituted with MCA1(1-173)D21N-6H or MCA2(1-173)D21N-6H. $Ca^{2+}$ and a detergent were added as described above. The mean and standard deviation of three measurements. In **a**, the p-values of the two-sided t-test between the data from MCA1(1-173)-6H or MCA2(1-173)-6H and those from control liposomes are indicated at 2.5 mM $CaCl_2$.

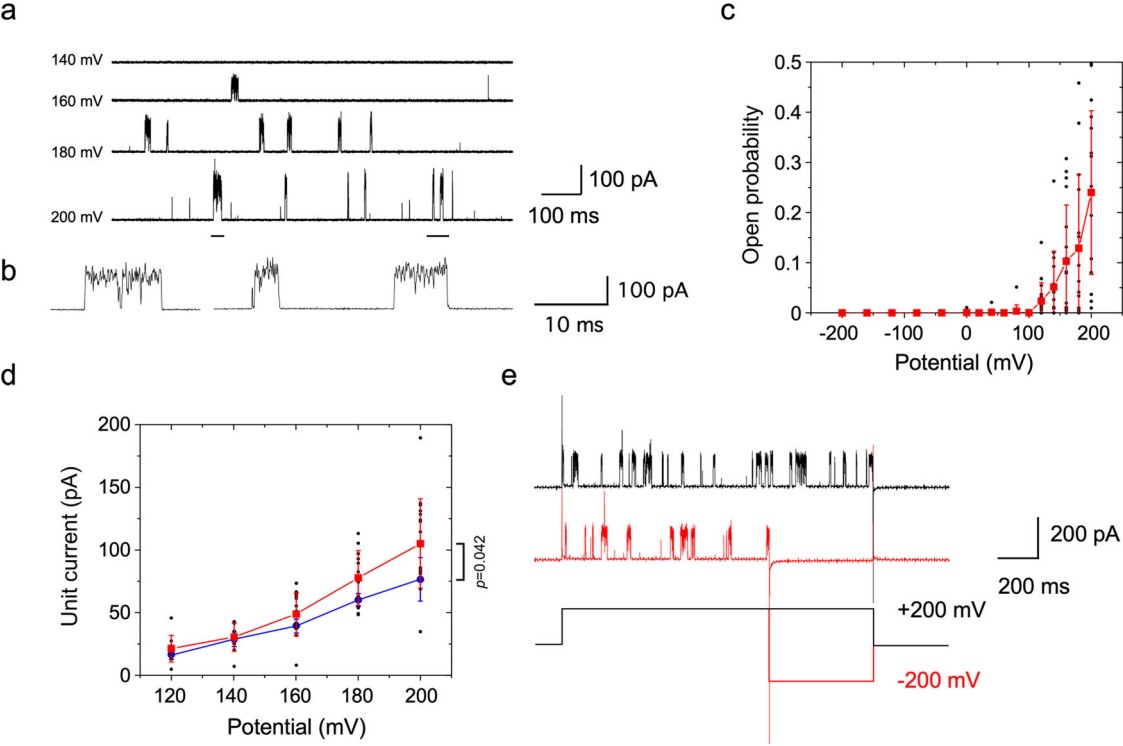

**Fig. 4 Voltage activation of MCA2(1-173) reconstituted in the lipid bilayer examined with the patch clamp technique. a** Channel current at a holding potential of 140 to 200 mV. Potential values indicate the pipette potential with respect to the bath potential. **b** Two enlarged traces of the currents corresponding to those underlined at 200 mV shown in **a**. **c** Changes in the open probability with the holding potential. **d** Changes in unit current with the holding potential in the presence of 200 mM KCl (red) or potassium gluconate (blue) in the bath solution. **e** Channel activity at a positive potential followed by the application of a positive (black) or negative potential (red). In **c** and **d**, the mean and standard deviation are shown. Small circles indicate individual data for **c** (KCl) and **d**. Data from 17 (**c**, **d**: KCl) and 6 (**d**: potassium gluconate) patches. In **d**, the *p*-value of the two-sided *t*-test between the data obtained in the presence of KCl and those obtained in the presence of potassium gluconate at +200 mV is indicated.

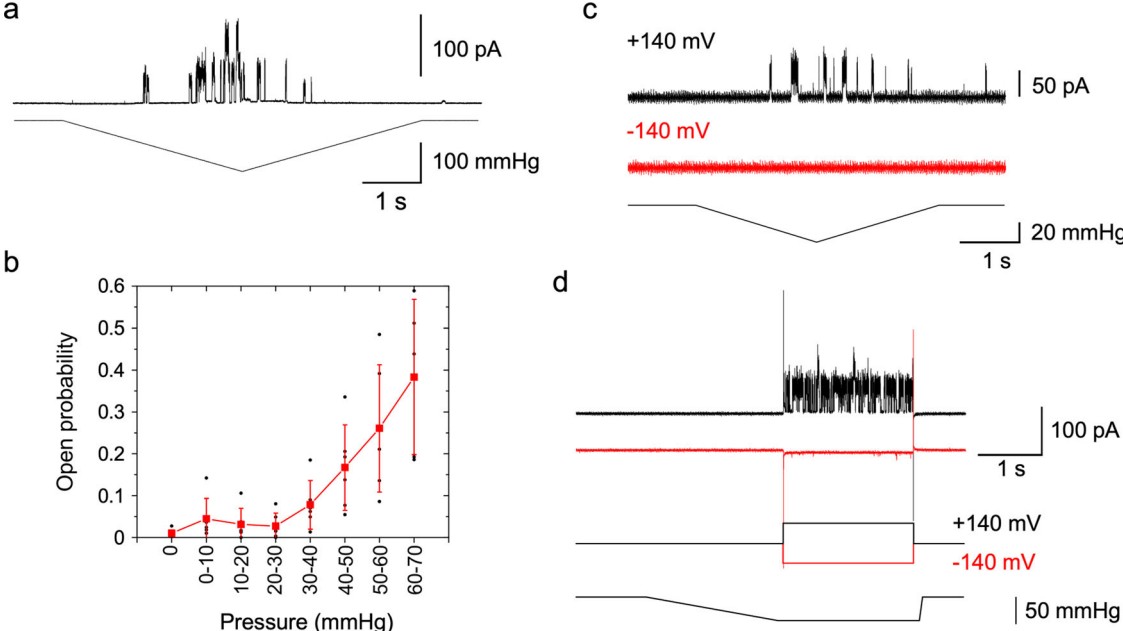

**Fig. 5 Stretch activation of MCA2(1-173) reconstituted in the lipid bilayer. a** Channel current (top trace) when a negative pressure (lower trace) was applied. The holding potential was 100 mV. **b** Changes in the open probability with the application of negative pressure. Data collected at 140 and 160 mV were combined. The mean and standard deviation are shown in red. Small circles indicate individual data. Data from 6 patches. **c** Channel current when the same pressure protocol was applied at +140 mV (black) or −140 mV (red) to a single patch. **d** Channel current when a positive (black) or negative potential (red) step was applied in the presence of negative pressure.

Therefore, we failed to measure the reversal potential because of strict rectification or steep voltage dependence of open probability.

**C-terminally truncated MCA2 is inherently mechanosensitive**. The mechanosensitivity of MCA2(1-173) was examined by applying negative pressure through the patch pipette. Channel activity was not detected in the absence of pressure at +100 mV; however, when negative pressure was gradually increased, channel events were initiated (Fig. 5a). Channel opening occurred more frequently at a high pressure, while the resting level of activity returned when pressure was released. Pressure dependence was obtained by sectioning the current trace according to the pressure range with a bin of 10 mmHg. Open probability markedly increased at pressures between 30 and 70 mmHg (Fig. 5b). Reaching saturation of the open probability at higher pressures were not possible due to patch breakage. Since the observed current events had discrete and well-defined closed- and open-state levels and open probabilities followed the beginning of sigmoid dose-response curves on voltage and pressure, MCA2(1-173) appears to be a *bona fide* tension- and voltage-activated channel.

We then investigated whether MCA2(1-173) is activated by negative pressure at negative potentials. When negative pressure was applied on the same membrane patch at a positive or negative potential, channel opening was observed at a positive potential, but not at a negative potential (Fig. 5c). Since the channel may have gradually entered an inactive state at a negative potential, negative pressure was applied at 0 mV and then followed by a positive or negative potential. Channel activation again occurred at a positive potential, but not at a negative potential (Fig. 5d).

These results indicate that the channel activity of MCA2(1-173) depended on voltage and membrane stretch. We then examined the relationship between voltage and stretch activation. We initially tested gadolinium, which affects MS channel activation by changing the physical properties of the lipid bilayer[38]. When gadolinium was present in the bath solution, pressure application did not markedly increase channel activity over that in the absence of pressure (Fig. 6a). Pressure dependence confirmed that gadolinium suppressed activation by pressure at concentrations of 10 and 100 μM (Fig. 6b). In contrast, voltage activation was largely unaffected by gadolinium (Fig. 6c). We then examined the effects of GsMT4, a spider venom toxin that inhibits cationic mechanosensitive channels[39]. GsMT4 reduces the effective magnitude of membrane stretch by penetrating deeper into the lipid monolayer upon membrane stretch[40]. GsMT4 also affected stretch activation, but not voltage activation (Fig. 6d, e). Therefore, the activation of MCA2(1-173) by tension is likely blocked independently of the activation by voltage. The blockage of activation by tension appears to occur through a modification of the bilayer lateral pressure imposed by these MS channel-specific blockers.

MS channels that are activated by membrane tension may also be activated by altering the intrinsic membrane curvature[41,42]. We applied lysophosphatidylcholine (LPC), an inverted cone-shaped lipid, to the patch membrane reconstituted with MCA2(1-173) and found that channel activity increased without the application of pressure (Fig. 6f). Open probability was 0.11 ± 0.10 before the application of LPC and increased to 0.53 ± 0.16 just before the membrane patch broke (such as the time shown as the bar in Fig. 6f, n = 5).

## Discussion

The present results showing an increase in channel activity with pressure, inhibition by MS channel blockers, and activation by curvature and area stresses asymmetrically imposed by lysolipids

support MCA2(1-173) functioning as a MS channel that is directly activated by membrane tension without additional components, such as a cytoskeleton or extracellular matrix. These characteristics also indicate that current did not arise from the destabilization of the proteoliposome membrane. Direct activation by membrane tension has also been reported for other MS channels[30–33]; nevertheless, MCA2(1-173) may represent the simplest form of an MS channel. The pore region of MCA2(1-173) may be similar to that formed by full-length MCA2, which is supported by the present result showing that MCA2(1-173) formed a tetramer in DDM micelles. Since the transmembrane α helix of MCA2 is amphipathic, the hydrophilic side including Asp-21 may assemble to form a channel pore and the hydrophobic side interacts with membrane lipids[17]. The failure to detect the influx of $Ca^{2+}$ in liposomes reconstituted with MCA2(1-173)D21N supports this view.

In the absence of tension, MCA2(1-173) was activated by a voltage of 120 mV or higher, which we assume corresponds to a membrane potential of −120 mV or lower in vivo. This threshold is above the resting potential recorded in the mesophyll cells (−160 to −180 mV) and root cells (−180 mV) of Arabidopsis[43,44]. However, the spontaneous opening of MCA2(1-173) at the resting potential is unlikely to occur because MCA2(1-173) allows the passage of $Ca^{2+}$, which activates various targets. We speculate that the threshold is more negative or voltage activation is suppressed in full-length MCA2 or in vivo. Since pressure activated MCA2(1-173) at voltages at which voltage activation did not occur, a natural key stimulus to open MCA2(1-173) may be membrane tension.

The absence of channel currents at negative holding potentials in MCA2(1-173) indicates strong rectification in which current flows in one direction only. This absence may also be attributed to steep voltage dependence of the open probability or activation only at positive potentials. The mechanism for rectification has been extensively examined in inward rectifier potassium channels. Rectification in inward rectifiers is not the result of an inherent property of a channel itself. It is due to channel blockade by cations, such as $Mg^{2+}$ and polyamines[45]. $Mg^{2+}$ in the patch clamp solution used in the present study may have contributed to rectification. The observation of rectification also indicates that MCA2(1-173) proteins were incorporated into the liposome lipid bilayer in a unidirectional orientation and not in a random fifty-fifty manner. Their directional incorporation into liposomes is not unexpected because membrane proteins, including ion channels, are incorporated directionally when the small, curved liposomes are initially formed and the dehydration/rehydration method is then applied for proteoliposome preparations. For example, the bacterial MS channel, MscS, is incorporated in a 100% right-side-out configuration[46]. Various ion channels, such as TREK-1, KirBac1.1, and TRPV1, show rectification when reconstituted into liposomes, indicating that these channels are incorporated into membranes in a unidirectional manner[47–49].

The detection of $Ca^{2+}$ flow in the fluorescence experiment indicates that MCA1(1-173) and MCA2(1-173) are permeable to $Ca^{2+}$. This permeability is consistent with the findings of an in vivo experiment using yeast cells expressing MCA1(1-173) and MCA2(1-173)[17]. On the other hand, the reduction in unitary channel current following the replacement of KCl with potassium gluconate implies the conduction of chloride ions. Therefore, MCA2(1-173) is probably an ion channel that is permeable to both cations and anions.

It currently remains unclear why $Ca^{2+}$ flowed into liposomes in fluorescence experiments in the absence of pressure and voltage. Since the majority of liposomes that form after 30 min of sonication have a diameter smaller than 100 nm[50], their curvature may be sufficient to activate MCA2(1-173) because changes in

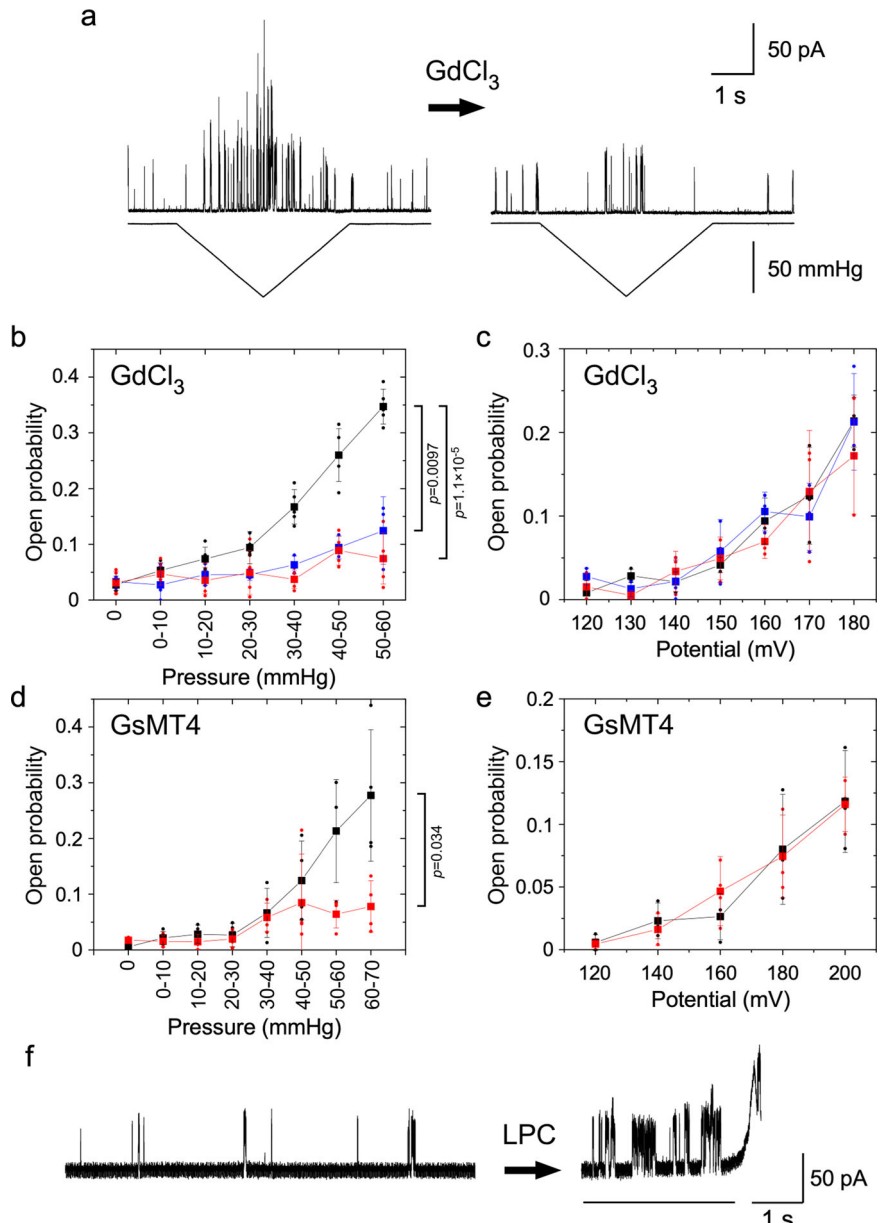

**Fig. 6 Effects of mechanosensitive channel blockers on channel activities of MCA2(1-173). a** Negative pressure was applied before (left) and after (right) the application of 30 μM GdCl$_3$. Current (top trace) and pressure (lower trace) are shown. The holding potential was 140 mV. **b, c** Changes in the open probability with pressure (**b**) and voltage (**c**) when 0 (black), 10 (blue), or 100 (red) μM GdCl$_3$ was applied. **d, e** Changes in the open probability with pressure (**d**) and voltage (**e**) when 0 (black) or 5 (red) μM GsMT4 was applied. **f** Current traces before (left) and after (right) the application of LPC. The membrane was broken at the end of recording. The holding potential was 140 mV. In **b, c, d**, and **e**, the mean and standard deviation are shown as squares and bars. Small circles indicate individual data. Data from 5 (**b**), 3 (**c**), 4 (**d**), and 3 (**e**) patches. In **b** and **d**, the p-values of the two-sided t-test between the data obtained in the presence of blockers and those obtained in their absence are indicated.

membrane curvature induced by the application of LPC activated MCA2(1-173) (Fig. 6f).

In summary, we herein demonstrated that MCA2(1-173) proteins synthesized with a cell-free expression system formed an MS channel in liposome membranes in vitro. Purified MCA1(1-173) proteins may exhibit similar activity because the two truncated proteins showed essentially the same activity in the in vitro assay system (Fig. 3) and in yeast cells[17]. This speculation is further supported by the amino acid sequence alignment of MCA1(1-173) showing 78% identity and 93% similarity without gaps to MCA2(1-173). However, it should be remembered that even a single amino acid substitution may cause a change in the function of proteins. In this context, further electrophysiological studies on MCA1(1-173) are warranted to compare the properties of MCA1(1-173) and MCA2(1-173). We chose MCA2(1-173) first because a three-dimensional structural model of MCA2 has been reported[18]. Nevertheless, this is the first study to demonstrate that plant-specific membrane proteins self-assemble to become an active ion channel. Consequently, the coiled-coil and PLAC8 motifs, which are absent in MCA1(1-173) and MCA2(1-173), do not appear to be necessary for the formation or activity of MS channels and, thus, may be involved in the cellular modulation of MCA channels because they are present on the cytoplasmic side[18]. The present study will serve as a starting point for obtaining a more detailed understanding of plant MS channels in terms of structural and regulatory characteristics.

## Methods

**In vitro synthesis and purification of MCA proteins**. MCA1, MCA2, and their derivatives were prepared as follows. We herein explain the procedure for the MCA2(1-173) protein as a representative for simplicity. MCA2(1-173) fused C-terminally with a 6 × His tag (-MGSHHHHHH), designated MCA2(1-173)-6H, was synthesized in vitro using the ProteoLiposome Expression Kit (CellFree Sciences Co., Ltd., No. CFS-TRI-PLE, Ehime, Japan) according to the manufacturer's protocol. A DNA fragment encoding MCA2(1-173)-6H was inserted between the *Eco*RV and *Not*I restriction sites of the in vitro expression vector, pEU-E01-MCS (CellFree Sciences Co., Ltd.). A translation reaction was conducted at 15 °C for 20 h in the presence of asolectin liposomes. The crude ProteoLiposome fraction containing synthesized MCA2(1-173)-6H proteins was isolated by centrifugation at 20,000 × g at 4 °C for 10 min, and washed four times with PBS by centrifugation. The final precipitate of the PBS wash from a 1-ml translation reaction mixture was suspended in 1.5 ml of solubilization buffer (50 mM sodium phosphate buffer, pH 8.0, 0.3 M NaCl, 1% DDM, and 50 mM imidazole) and centrifuged as described above. A portion (0.5 ml) of the supernatant (a total of 1.5 ml) was applied three times to a Ni-NTA Spin column (QIAGEN, No. 31014, Hilden, Germany), and the column was washed with 0.6 ml of solubilization buffer four times. MCA2(1-173)-6H proteins were eluted from the column with 0.6 ml of elution buffer (50 mM sodium phosphate buffer, pH 8.0, 0.3 M NaCl, 1% DDM, and 0.3 M imidazole), and the buffer was changed to MOPS buffer (5 mM MOPS-KOH, pH 7.2, 0.2 M KCl, and 1% DDM) using a PD-10 column (GE Healthcare Japan, No. 17-0851-01, Tokyo, Japan). The amount and purity of proteins were assessed by SDS-PAGE followed by silver staining. A total of 60 ~ 80 μg of the pure MCA2(1-173)-6H protein was typically obtained from a 1-ml translation reaction mixture.

The same procedure was applied to the purification of MCA1(1-173)-6H, MCA1(1-173)D21N-6H, and MCA2(1-173)D21N-6H.

**Cross-linking of MCA proteins using DSS**. To examine the multimerization of MCA1(1-173)-6H, MCA1(1-173)D21N-6H, MCA2(1-173)-6H, and MCA2(1-173)D21N-6H, each of the four proteins was synthesized and purified as described above and cross-linked with DSS (Thermo Fisher Scientific, Rockford, IL, USA) as previously reported[36]. Briefly, the Ni-NTA Spin column eluate containing one of the four proteins was applied to a PD-10 column to replace the elution buffer with DSS-reaction buffer (30 mM sodium phosphate, pH 7.5, 100 mM NaCl, and 1% DDM). DSS dissolved in dimethyl sulfoxide was added to the eluate from the PD-10 column to give final concentrations of 0.1, 0.3, and 1.0 mM, and the mixture was incubated at room temperature for 1 h. The reaction was stopped by adding 1 M Tris-HCl, pH 8.0 (a final concentration of 50 mM). After a 15-min incubation, the reaction mixture was subjected to SDS-PAGE followed by silver staining.

**Measurement of Ca$^{2+}$ influx into liposomes**. Each of the purified MCA1(1-173)-6H, MCA2(1-173)-6H, MCA1(1-173)D21N-6H, and MCA2(1-173)D21N-6H proteins were reconstituted into liposomes as previously described[36,51]. Twenty milligrams of L-α-phosphatidylcholine (P5638, Sigma-Aldrich) was dissolved in 0.5 mL of chloroform and the film was dried for one hour onto a glass test tube with nitrogen gas. The lipid film was kept in a vacuum for more than one hour. One milliliter of dehydration/rehydration (D/R) buffer (200 mM KCl and 5 mM MOPS-KOH, pH 7.2) was added and the suspension was vortexed until no particles were visible. The suspension was clarified with a bath sonicator at 30 °C for 10 min.

The channel preparation was mixed with 9.2 mg lipid at a protein-lipid ratio of 1:1,000. The mixture was supplemented with EGTA-KOH, pH 7.4 (1 mM final concentration), n-octyl-β-D-glucoside (1%), and Fluo-4 (0.03 μM). Approximately 400 mg of Bio-Beads SM-2 Adsorbents (BioRad, Hercules, CA, USA) prehydrated in wash buffer (D/R buffer supplemented with 1 mM EGTA-KOH, pH 7.4) was added to the sample and the mixture was incubated for 3 h. The liquid fraction was taken and incubated with fresh Bio-Beads overnight. The fluorescent dye that was not incorporated into liposomes was removed by exchanging the buffer with wash buffer using a desalting column (Zeba Spin Desalting Column, 7 K MWCO, Thermo Scientific, Rockford, IL, USA).

The fluorescence of Fluo-4 incorporated into liposomes was measured using a fluorescence spectrometer (PerkinElmer LS55, Waltham, MA, USA). Excitation and emission wavelengths were 495 and 527 nm, respectively (slit width: 5 nm). Ca$^{2+}$ concentrations were increased by the sequential addition of CaCl$_2$ solution. Liposomes were lysed by adding 1% (final concentration) n-octyl-β-D-glucoside at the end of measurements to obtain a measure of the total amount of Fluo-4 and normalize measurements.

**Patch clamp experiments on liposomes**. A lipid suspension was prepared as described above. The channel preparation was mixed with 10 mg lipid at a protein-lipid ratio of 1:1,000. n-Octyl-β-D-glucose was added to yield a final concentration of 1%. The sample was incubated on a seesaw shaker at 4 °C for one hour. Approximately 400 mg of Bio-Beads prehydrated in D/R buffer was added to the sample and incubated for 3 h. The liquid fraction was taken and incubated with fresh Bio-Beads overnight. The liquid fraction was ultracentrifuged at ~160,000 × g for 60 min. The pellet was suspended in 50 μl dehydration buffer consisting of 10 mM MOPS-KOH, pH 7.2, and 5% ethylene glycol. The suspension was spotted

on an 8-well printed slide and dehydrated under a vacuum at 4 °C overnight. Between 10 and 20 μl of D/R buffer was placed onto the spots, which were kept in a moist chamber for more than one hour. The rehydrated preparation was placed in patch clamp buffer (200 mM KCl, 90 mM MgCl$_2$, 10 mM CaCl$_2$, and 5 mM Hepes, pH 6.0) and allowed to form blisters. In assessments of ion selectivity, 200 mM KCl was replaced with 200 mM potassium gluconate in the bath solution.

The blister membrane was caught onto a glass pipette and excised from the blister. The pipette contained the same buffer as the bath solution . The holding potential was controlled and the current was amplified with a patch clamp amplifier (Axopatch 200B, Axon Instruments, Union city, CA, USA). Current recordings were filtered at 2 kHz and digitized at 5 kHz using a Digidata 1322 A interface with pCLAMP 9 software (Axon Instruments). Negative pressure was applied to the patch membrane through the pipette using a High-Speed Pressure Clamp-1 apparatus (HSPC-1; ALA Scientific Instruments, Westbury, NY, USA).

**Statistical analysis**. A two-sided *t*-test was used for statistical analyses. Equal variance was not assumed.

**Reporting summary**. Further information on research design is available in the Nature Research Reporting Summary linked to this article.

## Data availability

Data that support all experimental results in the present study are available within this manuscript and Figures. Source data for Figs. 1b, c, 2, 3, 4c, d, 5b, and 6b–e are provided as a Source Data file. Source data are provided with this paper.

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

## Acknowledgements

This work was supported by a Grant-in-Aid for Scientific Research on Innovative Areas (25120708 to H.I.) from the Ministry of Education, Culture, Sports, Science and Technology and a Grant-in-Aid for Scientific Research (C) (17K07370 to K.Y.) from the Japan Society for Promotion of Science, and AMED/PRIME (JP18gm5810013 to K.Y.) from the Japan Agency for Medical Research and Development. We are grateful to Ms. Tamiko Yoshioka for her financial assistance to H.I. We also thank Ms. Aki Nakamura, Dr. Megumi Yoshida, and Mr. K. Kurihashi for their technical assistance.

## Author contributions

H.I. conceived the research. K.Y., K.I., and H.I. designed the experiments. K.I. performed the experiments on protein synthesis and purification. K.Y. performed the experiments on fluorescence measurements and patch-clamp analyses. K.Y., K.I. and H.I. analyzed the data and wrote the manuscript. All authors read and approved the final manuscript.

## Competing interests

The authors declare no competing interests.
