## [Peer Review File · Nature Communications]

MCA_s in Arabidopsis are Ca²⁺-permeable mechanosensitive channels inherently sensitive to membrane tensionREVIEWER COMMENTS

Reviewer #1 (Remarks to the Author):

Comments on the manuscript "MCAs in Arabidopsis are Ca²⁺-permeable mechanosensitive channels innately sensitive to membrane tension" by Yoshimura et al.

This study provide clear and convincing results showing for the first time that MCAs are indeed mechanosensitive channels. The detailed electrophysiological characterization of MCA2(1-173) demonstrate that the channel is voltage dependent and directly activated by member tension. By a pharmacological approach, the authors shown that the channel is selectively affected (gadolinium and spider venom) in stretch but not in voltage activation. To complete the demonstration the authors also shown that altering membrane curvature with cone-shape lipid activated MCA2.

I have only a slight reproach to address to the authors. Based on fluorescent measurement performed on liposomes equipped with MCA1(1-173) and MCA2(1-173) it is deduced that MCAs are calcium permeable. In fact, Ca permeability should be assessed using patch clamp technique. By varying Ca⁺⁺ concentration of the bath and comparing reversal potential of the channel with the equilibrium potential of Ca⁺⁺ the authors will be able to precise the real Ca selectivity of MCA1. Indeed, looking at the bath and pipette solution composition KCl 200 mM, MgCl₂ 90 and CaCl₂ 10 mM (symmetrical conditions) the channel current could be dominated by Mg⁺⁺ or Cl⁻ fluxes (Cl⁻ outward). I precise, whatever the selectivity of the channel is, this study is well conducted and bring key information on MCA channels.

In Figure 4 f (LPC effect), the number of recordings should be precise and the Open probability in absence/presence of LPC should provided.

Minor comments

Page 4 line 19: MCA1-6H (1 is missing)

Page 7 line 5: "square wave" sound strange for me! Do the authors means step of current of constant amplitude?

Page 9, last line: Lu et al. 2004 should not appear in the text.

Reference 35 needs to be completed

Conclusion:

More than one decade ago, the authors discovered MCA proteins (PNAS 2007) and evidence their role in plant mechanosensing. Now, this paper bring the missing link providing the complete history of MCAs adding a significant contribution to the field of mechanosensing.

Reviewer #2 (Remarks to the Author):

The manuscript "MCAs in Arabidopsis are Ca²⁺ - permeable mechanosensitive channels...." By Yoshimura and co-workers presents an informative and important contribution to our understanding of plant mechanosensation.

The paper shows that the in-vitro expressed N-terminal fragments of MCA1 and MCA2 form functional channels in reconstituted liposomes, which can be activated by voltage and membrane tension. The work presents a minimal set of experiments that shows that MCA channels are inherently tension-sensitive and do not require any other component besides proper reconstitution to the lipid bilayer.

I have no technical critiques; the work is done in the best traditions of the channel-reconstitution field. Here are my suggestions on how to improve the readability of the paper.

1. In the introduction, I would pay more attention to the location and context in which these two plant-specific channels function in-vivo. Gravitropic responses in several locations in plants involve dense plastids sedimenting to the bottom of the cell which exerts some perturbing force directed normally to the membrane. It would be pertinent to insert a short paragraph describing the native setting where the MCA channels work.

2. Strategically, it would be important to compare the activities of reconstituted MCA (1-173) with activities of native full-length channels expressed in plant protoplasts. These channels can be potentially overexpressed in a callus tissue culture, which then can be converted to protoplasts. This comparison is needed because the 1-173 pore-forming fragment might exhibit only a part of channel properties (activation by tension and extreme voltage), whereas the rest of the protein may sensitize or modulate the response. For this short communication, the presented dataset seems sufficient as the first and

partial characterization of this novel class of channel proteins. The suggested in-situ protoplast recordings are optional.

3. Focusing on the 1-173 fragment, the authors should provide some minimal predictions of its structure, which part crosses the membrane and where is that critical D21 required for function. Also, it would be helpful to emphasize the difference between MCA1 and MCA2.

4. The requirement of high transmembrane potential to activate MCA channels by moderate tension predicts that the cells must be strongly energized. The biological function of MCA proteins in vivo, therefore, should be tested at different membrane potentials.

Fig. 1 can be omitted and the lane with the pure band can be incorporated into another figure showing functional data. It would be also nice to present an enlarged segment of a single-channel trace recorded with a higher time resolution to show possible subconductive states.

The manuscript requires English editing.

Reviewer #3 (Remarks to the Author):

In the manuscript "MCAs in Arabidopsis are Ca²⁺-permeable mechanosensitive channels innately sensitive to membrane tension" the authors explore the ability of the plant Mid1-

Complementing Activity (MCA) proteins 1 and 2 to directly respond to force from the lipids. They synthesize a truncated version of the channels in vitro and evaluate their activity by Ca⁺⁺ fluorescence and electrophysiology experiments. The authors report a pressure modulated activity in proteo-liposomes due to MCA2 truncated protein but there are several experimental controls that are needed to reach that conclusion.

1- Western blots are only shown for MCA2(1-173)-6H protein synthesis and purification fractions. The authors should show the other constructs' data as well (even as supplemental).

2- Western blots of the samples of the proteo-liposomes used for Ca⁺⁺ fluorescence measurements and electrophysiology need to be shown for all constructs to determine if the proteins are actually being incorporated into the lipids, and if there are differences in incorporation into the lipids between them. Additionally, it is important to see if the proteins can be detected in higher oligomeric states to support the conclusion that the truncated proteins self-assemble to become active ion channels.

This is of particular importance for the present work, since a new method (in vitro protein synthesis) and new mutants are used. Note that the MCA1, 2 D21N (1-173) mutants used in this paper were never studied before, since this mutation was only evaluated in the full-length channel (ref 17). Without these controls the data is difficult to interpret.

3- The N of the experiments for the results shown in figure 2 is missing. Also note that the fluorescence value at the 2.5mM Ca⁺⁺ point for the liposome (control) for figure 2B, is almost as high as the value of the MCA1,2 (1-173) containing liposomes in figure 2A. The authors should address the variability among experiments.

4- Only one lipid:protein ratio (1:1000) is shown. It would be interesting to see if there is any relationship between Ca⁺⁺ flux and/or channel activities at different lipid:protein ratios.

5- The frequency in which MCA2(1-173) activity was detected in patch needs to be reported (number of patches in which channel activity was observed/total number of patches assayed).

6- The voltage required to observe channel activity seems too high, especially for azolectin proteo-liposomes that tend to destabilize at high voltages. In addition, this observation is in sharp contrast with the authors explanation that the curvature of the liposomes alone will be enough to gate the channels in the Ca⁺⁺ fluorescence experiments.

7- The unit current data in Figure 3 c shows a very large variance, that needs to be addressed.

A point-by-point response to the reviewers' comments (written in red)

<Reviewer #1>

Comment 1: I have only a slight reproach to address to the authors. Based on fluorescent measurement performed on liposomes equipped with MCA1(1-173) and MCA2(1-173) it is deduced that MCAs are calcium permeable. In fact, Ca permeability should be assessed using patch clamp technique. By varying Ca^{++} concentration of the bath and comparing reversal potential of the channel with the equilibrium potential of Ca^{++} the authors will be able to precise the real Ca selectivity of MCA1. Indeed, looking at the bath and pipette solution composition KCl 200 mM, MgCl_2 90 and CaCl_2 10 mM (symmetrical conditions) the channel current could be dominated by Mg^{++} or Cl^- fluxes (Cl^- outward). I precise, whatever the selectivity of the channel is, this study is well conducted and bring key information on MCA channels.

Response 1: Due to the physiological significance of Ca^{2+} , it is important to demonstrate its permeability. Ca^{2+} permeability is generally evaluated by the reversal potential, at which a current-voltage curve crosses the voltage axis. This intersection, or reversal in current, was not observed in MCA2(1-173) because of strong rectification (Fig. 3a, c). A channel current was only generated at positive potentials and was absent at negative potentials (Fig. 3c). Therefore, it was not possible to assess reversal potential in MCA2(1-173). This explanation has been added to line 169-172.

As an alternative method to demonstrate Ca^{2+} permeability, we measured the signal from a Ca^{2+} -sensitive fluorescent dye and found that MCA1(1-173) and MCA2(1-173) were both permeable to Ca^{2+} (Fig. 2).

Comment 2: In Figure 4 f (LPC effect), the number of recordings should be precise and the Open probability in absence/presence of LPC should be provided.

Response 2: The number of recordings has been stated on line 200-201. The open probability of the traces shown in Fig. 4f has been presented on line 199-200. We did not provide statistical data because the open probability increased with time, possibly due to an increase in the amount of lysophosphatidylcholine inserted.

Minor comments

Comment 3: Page 4 line 19: MCA1-6H (1 is missing)

Response 3: This typo has been corrected (line 90).

Comment 4: Page 7 line 5: “square wave” sound strange for me! Do the authors means step of current of constant amplitude?

Response 4: We changed the phrase to “current steps with a constant amplitude” (line 156).

Comment 5: Page 9, last line: Lu et al. 2004 should not appear in the text.

Response 5: This reference has been removed from the revised manuscript.

Comment 6: Reference 35 needs to be completed.

Response 6: This reference (ref. 36 in the revised manuscript) has been fully described (line 463-464).

<Reviewer #2>

Comment 1: In the introduction, I would pay more attention to the location and context in which these two plant-specific channels function in-vivo. Gravitropic responses in several locations in plants involve dense plastids sedimenting to the bottom of the cell which exerts some perturbing force directed normally to the membrane. It would be pertinent to insert a short paragraph describing the native setting where the MCA channels work.

Response 1: According to this comment, we inserted a short paragraph describing the localization of MCA1 and MCA2 *in planta* (line 59-71). Consequently, we cited a new reference reporting their specific expression patterns (ref. 26).

Comment 2: Strategically, it would be important to compare the activities of reconstituted MCA (1-173) with activities of native full-length channels expressed in plant protoplasts. These channels can be potentially overexpressed in a callus tissue culture, which then can be converted to protoplasts. This comparison is needed because the 1-173 pore-forming

fragment might exhibit only a part of channel properties (activation by tension and extreme voltage), whereas the rest of the protein may sensitize or modulate the response. For this short communication, the presented dataset seems sufficient as the first and partial characterization of this novel class of channel proteins. The suggested in-situ protoplast recordings are optional.

Response 2: Thank you for suggesting a significant experiment using callus protoplasts. We understand that this experiment would reinforce the current *in vitro* study. However, we would like to avoid conducting this experiment currently because it is challenging and, therefore, takes time.

A possible role for the coiled-coil and PLAC8 motifs has been described in our manuscript (page 10, line 22-page 11, line, 1 of the original manuscript and line 256-260 of the revised manuscript).

Comment 3: Focusing on the 1-173 fragment, the authors should provide some minimal predictions of its structure, which part crosses the membrane and where is that critical D21 required for function. Also, it would be helpful to emphasize the difference between MCA1 and MCA2.

Response 3: Thank you for your suggestion. We added a schematic diagram showing the full-length and truncated forms (1-173) of MCA1 and MCA2 to Fig. 1a.

We did not obtain any data to show the functional difference between full-length MCA1 and MCA2. For example, Ca^{2+} influx activity in yeast cells (ref. 26) and membrane-stretch-induced cation currents in *Xenopus oocytes* (ref. 8) are essentially the same. Based on the identity (78%) and similarity (87%) in the amino acid sequence alignment and Ca^{2+} influx activity in yeast cells (ref. 17), we speculate that MCA1(1-173) and MCA2(1-173) have essentially the same electrophysiological characteristics. We briefly explained this in the revised manuscript (line 253-254).

Comment 4: The requirement of high transmembrane potential to activate MCA channels by moderate tension predicts that the cells must be strongly energized. The biological function of MCA proteins *in vivo*, therefore, should be tested at different membrane potentials.

Response 4: Plants have a significantly larger negative resting potential than animals. The resting potential in *Arabidopsis* is -160 to -180 mV in mesophyll cells and -180 mV in root cells (line 219-221). Therefore, we consider the

transmembrane potentials used in the present study to be within the physiological range.

Comment 5: Fig. 1 can be omitted and the lane with the pure band can be incorporated into another figure showing functional data. It would be also nice to present an enlarged segment of a single-channel trace recorded with a higher time resolution to show possible subconductive states.

Response 5: We are grateful for the comment on Fig. 1. However, in consideration of comment #3 that “Focusing on the 1-173 fragment, the authors should provide some minimal predictions of its structure”, we considered the combination of information on the 1-173 fragment and the purification steps of the fragment to be informative for readers and, thus, restructured Fig. 1.

Based on the comment made by this Reviewer, enlarged traces of single channel openings have been added as Fig. 3b. The trace did not show a stable subconducting state (line 158-159).

Comment 6: The manuscript requires English editing.

Response 6: The manuscript has been subjected to English editing.

<Reviewer #3>

Comment 1: Western blots are only shown for MCA2(1-173)-6H protein synthesis and purification fractions. The authors should show the other constructs' data as well (even as supplemental).

Response 1: Thank you for this comment. We added a photograph of the purified MCA1(1-173) and D21N proteins visualized by silver staining (not by Western blotting) to Fig. 1c.

Comment 2: Western blots of the samples of the proteo-liposomes used for Ca⁺⁺ fluorescence measurements and electrophysiology need to be shown for all constructs to determine if the proteins are actually being incorporated into the lipids, and if there are differences in incorporation into the lipids between them. Additionally, it is important to see if the proteins can be detected in higher oligomeric states to support the conclusion that the truncated proteins self-assemble to become active ion channels.

This is of particular importance for the present work, since a new method (in vitro protein synthesis) and new mutants are used. Note that the MCA1, 2 D21N (1-173) mutants used in this paper were never studied before, since this mutation was only evaluated in the full-length channel (ref 17). Without these controls the data is difficult to interpret.

Response 2: We are grateful for this comment. We understand that we need to demonstrate that the proteins used in the present study were actually incorporated into liposomes. As explained in the revised manuscript (line 137-148), since the MCA1 and MCA2 proteins and their derivatives undergo aggregation with various sizes *in vitro*, difficulties are associated with separating reconstituted liposomes and protein aggregates by differential centrifugation.

Irrespective of this difficulty, we detected stretch-induced currents in MCA2(1-173) liposomes because the patch-clamp method is sufficiently sensitive to detect channel currents generated even from a small number of proteins in the membrane. Under the same conditions, we did not detect currents in vector liposomes. Therefore, we are confident that the MCA1(1-173) and MCA2(1-173) proteins were actually incorporated into liposomes in our experiments.

Regarding MCA1(1-173)D21N and MCA2(1-173)D21N, which did not mediate an increase in Ca^{2+} concentrations under identical conditions to those used for wild-type MCA1(1-173) and MCA2(1-173), we consider both proteins to have been incorporated into liposomes because Asn is slightly more hydrophobic than Asp and, thus, the D21N mutation is not expected to interfere with the incorporation of the MCA1(1-173) and MCA2(1-173) proteins into liposomes.

We did not demonstrate that the MCA1(1-173) and MCA2(1-173) proteins formed an oligomeric structure in reconstituted liposomes. As described in the first paragraph of the response to this Reviewer's comment (#2), we were unable to separate reconstituted liposomes and protein aggregates, and, thus, were unable to perform the important experiments suggested. As pointed out by this Reviewer, full-length MCA1 and MCA2 have been shown to form a homotetramer in yeast and insect cells (refs. 17,18). Even though data showing the oligomeric structures of the MCA1(1-173) and MCA2(1-173) proteins in reconstituted liposomes are limited, we consider it possible to speculate that the truncated proteins self-assemble to become active ion channels because they exhibited activity to mediate the incorporation of Ca^{2+} into reconstituted

liposomes and generate currents in response to stretching of the liposome membrane. We briefly discussed this point in the Discussion section of the revised manuscript (line 210-217).

Comment 3: The N of the experiments for the results shown in figure 2 is missing. Also note that the fluorescence value at the 2.5mM Ca⁺⁺ point for the liposome (control) for figure 2B, is almost as high as the value of the MCA1,2 (1-173) containing liposomes in figure 2A. The authors should address the variability among experiments.

Response 3: The N of the measurement in Fig. 2 has been added (line 526-527). The fluorescence value for liposomes (control) (0.54) in Fig. 2b was lower than those of MCA1,2 (1-173)-containing liposomes (0.68 and 0.63) in Fig. 2a. The increase in fluorescence in control liposomes upon the addition of Ca²⁺ may have been due to a residual calcium indicator in the medium. We suspect that the extent to which the calcium indicator was removed differed from experiment to experiment.

Comment 4: Only one lipid:protein ratio (1:1000) is shown. It would be interesting to see if there is any relationship between Ca⁺⁺ flux and/or channel activities at different lipid:protein ratios.

Response 4: We attempted various protein:lipid ratios during the course of selecting experimental conditions, and found that a lower protein:lipid ratio markedly reduced the chance of channel observations. A ratio of 1:1000 was adopted because the patch often contained only one channel, which is advantageous for a channel analysis. We did not extensively examine a higher protein:lipid ratio because of the difficulties associated with obtaining a high yield, which may have been due to protein aggregation.

Comment 5: The frequency in which MCA2(1-173) activity was detected in patch needs to be reported (number of patches in which channel activity was observed/total number of patches assayed).

Response 5: We observed MCA2(1-173) activity in 59 out of the 124 patches examined. This information has been added to line 160.

Comment 6: The voltage required to observe channel activity seems too high, especially for azolectin proteo-liposomes that tend to destabilize at high voltages.

In addition, this observation is in sharp contrast with the authors explanation that the curvature of the liposomes alone will be enough to gate the channels in the Ca⁺⁺ fluorescence experiments.]

Response 6: In our experiments, the patch membrane was stable at voltages from -200 and +200 mV. The observed current did not appear to be due to a destabilized membrane because a channel-like current was not observed in negative control experiments. The observation of rectification, inhibition by channel blockers, and activation by lysophosphatidylcholine also supports the view that the current was not due to membrane breakage. We added a sentence referring to the possibility of membrane destabilization to the revised manuscript (line 208-209).

In patch experiments, membrane stretch activated the current at membrane potentials at which voltage alone did not (Fig. 3d), suggesting that a high voltage is not required for mechanical activation. Activation by the application of lysophosphatidylcholine also suggests that an increase in membrane curvature is sufficient to activate MCA2(1-173). Therefore, we considered the membrane curvature of small unilamellar liposomes in fluorescence experiments to have activated MCA2(1-173) because the typical radius of small unilamellar liposomes is as small as ~100 nm.

Comment 7: The unit current data in Figure 3 c shows a very large variance, that needs to be addressed.

Response 7: Apart from six outliers (5 and 46 pA at 120 mV, 7 pA at 140 mV, 8 pA at 160 mV, and 35 and 190 pA at 200 mV in Fig. 3d), 61 other measurements appeared to vary to a similar extent to that reported in previous studies on channels reconstituted into lipid bilayers (e.g. Fig. 3 in Koszela-Piotrowska et al., Cell Mol Biol. Lett. 12: 493-508, 2007; Fig. 5C in Pavlenok et al., PloS One 7: e38726, 2012). We did not remove the outliers due to the lack of a reasonable explanation to exclude them.

Concluded.

REVIEWER COMMENTS

Reviewer #1 (Remarks to the Author):

Comments on the revised manuscript "MCAs in Arabidopsis are Ca²⁺-permeable mechanosensitive channels inherently sensitive to membrane tension" by Yoshimura et al.

This new version of the manuscript is improved compared to the previous one.

(1)- Nevertheless I have to say that I was disappointed about the answer to my comment 1 concerning the calcium permeability of MCA. Indeed considering: (1) the high Cl⁻ concentration of 400 mM (bath/pipette), (2) the low Ca²⁺ concentration of 10 mM (bath/pipette), (3) the high amplitude of the single channel current of 100 pA at 200 mV (channel conductance about 500 pS), Chloride ions likely contribute to the current recorded. Even though a direct determination of reversal potential is not possible due to a strong rectification of MCA current, (i) a tail current protocol could be performed (steps of voltage during mechanical activation) to determine inversion potential (ii) or the study of the effect of reducing Cl⁻ concentration in one compartment (keeping cations constant) on the channel amplitude will bring basic information on anion/cation permeability of MCA.

It has been recently shown that a cytosolic calcium increase is observed during the activation of the anion permeant MscS-like mechanosensitive channel (Basu et al, 2020). This indicates that calcium measurement with a cytosolic probe is not sufficient to access channel permeability.

(2)- Comment 2: I believe that (fig. 4f, line 200-201) statistics should be provided even though with a large dispersion of the values

Conclusion: my first comment expresses more regret than an absolute requirement

Reviewer #2 (Remarks to the Author):

After re-reading the manuscript “MCAs in Arabidopsis are Ca²⁺-permeable mechanosensitive channels inherently sensitive to membrane tension” by Yoshimura and co-workers, I found that the authors did a thorough job revising it. The paper is informative and describes an important contribution.

I have only one principle point requiring clarification. The authors have shown that both MCA1(1-173)D21N and MCA2(1-173)D21N produce an undetectable Ca²⁺ influx in liposome assays. The absence of a critical carboxylic residue in the middle of the TM helix may abolish Ca²⁺-selectivity. But the question is whether these channels gate at all? Do these mutants gate under voltage or stretch in patch-clamp experiments?

The rest of my comments are just to improve the readability of the text. Please consider this.

Abstract, line 27: Blockers for MS channels inhibit activation by stretch, but not by voltage.

Lines 35-36: as well as intrinsic mechanical stress produced by cell division or cellular growth

Line 52: present in archaea, bacteria, protists...

Line 55: Out of the entire MCA1 protein chain comprising 421 residues (Fig. 1a), the N-terminal 173 residues...

Line 59: In Plantae, ... (Latin name of the kingdom)

Line 76: demonstrated to be inherently mechanosensitive, their counterparts in bacteria and animals have been.

Line 99: Please specify whether the new buffer had another (dialyzable) detergent.

Line 103: ... visible protein contaminants...

Line 134: When MCA1(1-173)D21N and MCA2(1-173)D21N mutants lacking a critical carboxylic residue in the pore-lining helices were reconstituted...

Line 145: Nevertheless, as Asn is less polar than Asp, ...

Line 152: The C-terminally truncated MCA2 is inherently mechanosensitive

Line 153: the patch-clamp technique

Lines 155-156: When +160 mV was applied to the patch membrane excised from a blister, sparse current steps with a fixed unitary amplitude were detected (Fig. 3a).

It would be also good to mention in the figure legend whether + and – voltages refer to pipette or bath.

Lines 158-159: Some flickering was observed at the upper level of unitary current events, but no long-lived subconducting states were visually detected (Fig. 3b). (I mean that a more thorough analysis would require amplitude histograms).

Lines 163-164: As current... - this phrase can be re-written and logically placed after Line 181.

Consider this: As the observed current events had discrete and well-defined closed- and open-state levels and the open probabilities followed the beginnings of sigmoid dose-response curves on voltage and pressure, MCA2(1-173) appears to be a bonafide tension- and voltage-activated channel.

Lines 171-172: the current occurred in one direction only and the current-to-voltage (I-V) curve could not be reliably extrapolated to zero voltage.

Line 191: In addition to Ref. 33 please reference this paper, which gives more insight on the GsMTx4 mechanism: Gnanasambandam R, et al. GsMTx4: Mechanism of Inhibiting Mechanosensitive Ion Channels. *Biophys J.* 2017, 112(1):31-45.

Lines 192-194: Therefore, the stretch activation of MCA2(1-173) can be blocked independently of voltage activation and appears to require a change in bilayer lateral pressure exerted by these MS channel-specific blockers (refs to Gd and GsMTx4 papers).

Line 205: MS channel blockers, and activation by curvature and area stresses asymmetrically imposed by lysolipids support MCA2(1-173) functioning as...

Line 235: ...and not in a random fifty-fifty manner...

Lines 237-238: ion channels are known to be incorporated directionally when the small curved liposomes are initially formed and then the dehydration/rehydration method is applied for proteoliposome preparations.

Line 249: In summary, we herein showed that MCA2(1-173) proteins synthesized and purified from a cell-free expression system... (I would not use 'artificially synthesized').

Reviewer #3 (Remarks to the Author):

There are more aspects of the protein purification and detection that need to be address by the authors before publication.

The point of the paper is to prove that the MCA channels are intrinsically mechanosensitive.

The main problem is that the in vitro synthesis method used, brings into question the proper folding of these transmembrane proteins. A truncated version of the MCA2 protein, generated by in vitro synthesis is being used to prove intrinsic channel mechano-sensitivity. The assumption is that a truncated MCA synthesized in vitro is functional like in its native form. But the authors acknowledge that the protein aggregates easily, so proper protein folding and heterodimerization are in question.

The following issues should be addressed before publication.

1- “As explained in the revised manuscript (line 137-148), since the MCA1 and MCA2 proteins and their derivatives undergo aggregation with various sizes in vitro, difficulties are associated with separating reconstituted liposomes and protein aggregates by differential centrifugation.”

The authors unsuccessfully tried to separate the proteo-liposomes from the protein aggregates by differential centrifugation. Have they tried to separate the protein aggregates from the protein incorporated into liposomes using a sephadex column? This method is often used to separate proteo-liposomes from free dyes.

2-The truncated MCA1 and 2 proteins in (ref 17) were detected with a Apep2 antibody at a 33 kDa marker. Here the band shown by silver staining is 19 kDa. Did the authors try the antibody used in the previous publication, Apep2? Does the protein undergo posttranslational modifications that can explain this big shift in size? A proper discussion of this issue should be included.

3-In the same reference 17 the authors cannot determine a plasma membrane localization of the MCA 1-173 mutants or the full-length MCA D21N mutant, they claim that most of the protein is in vacuoles. A discussion is needed.

This publication would greatly benefit from a direct prove of the oligomeric state of MCA2 when incorporate in the liposomes.

Point-by-point responses (written in red) to the Reviewers' comments are listed below.

Responses to Reviewer #1:

[Chloride ions likely contribute to the current recorded. Even though a direct determination of reversal potential is not possible due to a strong rectification of MCA current, (i) a tail current protocol could be performed (steps of voltage during mechanical activation) to determine inversion potential (ii) or the study of the effect of reducing Cl⁻ concentration in one compartment (keeping cations constant) on the channel amplitude will bring basic information on anion/cation permeability of MCA.]

[Response] (i) Thank you for the suggestion of a tail current protocol. We applied a positive or negative voltage step in the presence of negative pressure, and found that the channel current was activated at a positive potential, but not at a negative potential (Fig. 5d) (line 220-230). Transient activation was not observed. We also applied a positive or negative voltage step after preactivation at a positive voltage, but did not observe any channel activity at a negative potential (Fig. 4e) (line 197-209). Therefore, it was not possible to record current at a negative potential using the tail current protocol because of strong rectification.

(ii) We reduced the concentration of Cl⁻ in the bath solution by replacing KCl with potassium gluconate. Unit current was reduced by approximately 30%, as shown in Fig. 4d. This result implies that MCA2(1-173) is permeable to chloride ions at least in part (line 197-200). A discussion on ion selectivity has been added to the revised manuscript (line 295-301).

[It has been recently shown that a cytosolic calcium increase is observed during the activation of the anion permeant MscS-like mechanosensitive channel (Basu et al, 2020). This indicates that calcium measurement with a cytosolic probe is not sufficient to access channel permeability.]

[Response] Basu et al. (2020) suggested that the activation of the MSCL10 MscS-like mechanosensitive channel leads to downstream Ca^{2+} influx. The downstream signal transduction pathway was preserved because they used intact *Arabidopsis* cells. However, purified channel proteins and lipids were used to measure Ca^{2+} influx into liposomes in the present study. Therefore, we consider the Ca^{2+} increase detected by the fluorescent probe to be due to the influx of Ca^{2+} through the channel protein rather than the activation of the downstream pathway.

[(2)- Comment 2: I believe that (fig. 4f, line 200-201) statistics should be provided even though with a large dispersion of the values]

[Response] Statistical analyses of the channel open probability have been described on line 250-252.

Responses to Reviewer #2:

[The authors have shown that both MCA1(1-173)D21N and MCA2(1-173)D21N produce an undetectable Ca^{2+} influx in liposome assays. The absence of a critical carboxylic residue in the middle of the TM helix may abolish Ca^{2+} -selectivity. But the question is whether these channels gate at all? Do these mutants gate under voltage or stretch in

patch-clamp experiments?]

[Response] The structure and gating mechanism of MCA channels are the foci of our next study and we are currently performing a cryo-EM study. The position and function of Asp-21 are one of the main targets of this study. We appreciate the Reviewer's interest, but would like to answer this question in our future study.

We are grateful for the following comments that markedly improved the readability and preciseness of the manuscript.

[Abstract, line 27: Blockers for MS channels inhibit activation by stretch, but not by voltage.]

[Response] Rewritten as suggested (line 27).

[Lines 35-36: as well as intrinsic mechanical stress produced by cell division or cellular growth]

[Response] Rewritten as suggested (line 35).

[Line 52: present in archaea, bacteria, protists...]

[Response] Rewritten as suggested (line 52).

[Line 55: Out of the entire MCA1 protein chain comprising 421 residues (Fig. 1a), the N-

terminal 173 residues...]

[Response] Rewritten as suggested (line 55-57).

[Line 59: In Plantae, ... (Latin name of the kingdom)]

[Response] Rewritten as suggested (line 60).

[Line 76: demonstrated to be inherently mechanosensitive, their counterparts in bacteria and animals have been.]

[Response] Rewritten as suggested (line 77-78).

[Line 99: Please specify whether the new buffer had another (dialyzable) detergent.]

[Response] We described the composition of the buffer as “30 mM sodium phosphate, pH 7.5, 100 mM NaCl, 1% DDM” in the Methods section (line 357) and added “a DDM-containing buffer” to the Results section (line 100).

[Line 103: ... visible protein contaminants...]

[Response] Rewritten as suggested (line 113).

[Line 134: When MCA1(1-173)D21N and MCA2(1-173)D21N mutants lacking a critical carboxylic residue in the pore-lining helices were reconstituted...]

[Response] Rewritten as suggested (line 110).

[Line 145: Nevertheless, as Asn is less polar than Asp, ...]

[Response] Rewritten as suggested (line 174).

[Line 152: The C-terminally truncated MCA2 is inherently mechanosensitive]

[Response] Rewritten as suggested (line 211).

[Line 153: the patch-clamp technique]

[Response] Rewritten as suggested (line 180).

[Lines 155-156: When +160 mV was applied to the patch membrane excised from a blister, sparse current steps with a fixed unitary amplitude were detected (Fig. 3a).]

[Response] Rewritten as suggested (line 182-184).

[It would be also good to mention in the figure legend whether + and – voltages refer to pipette or bath.]

[Response] We added that the potential values used in the present study indicate pipette potentials (lines 182 and 640).

[Lines 158-159: Some flickering was observed at the upper level of unitary current events, but no long-lived subconducting states were visually detected (Fig. 3b). (I mean that a more thorough analysis would require amplitude histograms).]

[Response] Thank you for the suggestion of presenting an amplitude histogram. An all-point histogram of the open state trace has been included as Supplementary Fig. 1 and

referred to on line 187-188.

[Lines 163-164: As current... - this phrase can be re-written and logically placed after Line 181.]

Consider this: As the observed current events had discrete and well-defined closed- and open-state levels and the open probabilities followed the beginnings of sigmoid dose-response curves on voltage and pressure, MCA2(1-173) appears to be a bonafide tension- and voltage-activated channel.]

[Response] We are grateful for this suggestion. We replaced the original sentence with the suggested sentence and placed it at the position indicated (line 220-223).

[Lines 171-172: the current occurred in one direction only and the current-to-voltage (I-V) curve could not be reliably extrapolated to zero voltage.]

[Response] Rewritten as suggested (line 203-205).

[Line 191: In addition to Ref. 33 please reference this paper, which gives more insight on the GsMTx4 mechanism: Gnanasambandam R, et al. GsMTx4: Mechanism of Inhibiting Mechanosensitive Ion Channels. Biophys J. 2017, 112(1):31-45.

[Response] Thank you for this suggestion. The reference has been cited as ref. 42 and the text has been rewritten (line 240-241).

[Lines 192-194: Therefore, the stretch activation of MCA2(1-173) can be blocked independently of voltage activation and appears to require a change in bilayer lateral pressure exerted by these MS channel-specific blockers (refs to Gd and GsMTx4 papers).]

[Response] The text has been rewritten as “Therefore, the stretch activation of MCA2(1-173) may be blocked independently of voltage activation and appears to require an increase in bilayer lateral pressure, which is hindered by these MS channel-specific blockers.” (line 242-245).

Line 205: MS channel blockers, and activation by curvature and area stresses asymmetrically imposed by lysolipids support MCA2(1-173) functioning as...]

[Response] Rewritten as suggested (line 256-257).

[Line 235: ...and not in a random fifty-fifty manner...]

[Response] Rewritten as suggested (line 286-287).

[Lines 237-238: ion channels are known to be incorporated directionally when the small curved liposomes are initially formed and then the dehydration/rehydration method is applied for proteoliposome preparations.]

[Response] Thank you for this information. Rewritten as suggested (line 289-290).

[Line 249: In summary, we herein showed that MCA2(1-173) proteins synthesized and purified from a cell-free expression system... (I would not use ‘artificially synthesized’).]

[Response] Rewritten as suggested (line 307-308).

Reviewer #3 (Remarks to the Author):

There are more aspects of the protein purification and detection that need to be addressed by the authors before publication.

The point of the paper is to prove that the MCA channels are intrinsically mechanosensitive.

The main problem is that the in vitro synthesis method used, brings into question the proper folding of these transmembrane proteins. A truncated version of the MCA2 protein, generated by in vitro synthesis is being used to prove intrinsic channel mechanosensitivity. The assumption is that a truncated MCA synthesized in vitro is functional like in its native form. But the authors acknowledge that the protein aggregates easily, so proper protein folding and heterodimerization are in question.

The following issues should be addressed before publication.

1- “As explained in the revised manuscript (line 137-148), since the MCA1 and MCA2 proteins and their derivatives undergo aggregation with various sizes in vitro, difficulties are associated with separating reconstituted liposomes and protein aggregates by differential centrifugation.”

The authors unsuccessfully tried to separate the proteo-liposomes from the protein aggregates by differential centrifugation. Have they tried to separate the protein aggregates from the protein incorporated into liposomes using a sephadex column? This method is often used to separate proteo-liposomes from free dyes.

[Response] We performed preliminary experiments using blue dextran as a large molecule (~100 nm in diameter) and BSA as a small molecule (3-4 nm in diameter). We used Sepharose CL-2B, which is more suitable than Sephadex to separate large molecules, including liposomes (~100 nm in diameter). We found that although the two molecules were partially separated, separation was not sufficient despite the marked difference in

the sizes of the two molecules. Therefore, we anticipated difficulties with establishing a protocol for separating liposomes and the various sized aggregates of MCA proteins. We alternatively performed a cross-linking experiment on MCA proteins synthesized with a liposome-supplemented wheat germ cell-free translation system. We considered this method to enable us to show whether MCA proteins had assembled properly in the tetrameric form as a channel in the lipid bilayers of liposomes. Using DSS as a cross-linker, we detected the tetrameric forms of MCA1(1-173), MCA1(1-173)D21N, MCA2(1-173), and MCA2(1-173)D21N (see line 124-141, line 264-265, line 351-362, and Fig. 2). We consider these approaches to have answered the Reviewer's concerns regarding proper protein folding and oligomerization.

2-The truncated MCA1 and 2 proteins in (ref 17) were detected with a Apep2 antibody at a 33 kDa marker. Here the band shown by silver staining is 19 kDa. Did the authors try the antibody used in the previous publication, Apep2? Does the protein undergo posttranslational modifications that can explain this big shift in size? A proper discussion of this issue should be included.

[Response] Thank you for this comment based on a careful comparison between the molecular size (~33 kDa) of the truncated MCA proteins shown in ref. 17 and that (19-20 kDa) in this manuscript. We detected MCA1(1-173) and MCA2(1-173) with Apep2 antibodies that interact with the C-terminal region of both proteins. These bands corresponded to those identified by silver staining. Based on this result and the following observations, MCA1(1-173) and MCA2(1-173) did not undergo post-translational modifications.

In Fig. 7A of ref. 17, we avoided describing the HA4 tag (four tandem

hemagglutinin antigen repeats) fused to the C termini of MCA1(1-173) and MCA2(1-173). The molecular sizes of fused MCA1(1-173) and MCA2(1-173) were calculated to be 27.8 and 27.5 kDa, respectively, while those of MCA1(1-173)-6H and MCA2(1-173)-6H were 21.7 and 21.4 kDa, respectively. We explained this in the legend of Fig. 1 in the revised manuscript (line 615-617).

3-In the same reference 17 the authors cannot determine a plasma membrane localization of the MCA 1-173 mutants or the full-length MCA D21N mutant, they claim that most of the protein is in vacuoles. A discussion is needed.

[Response] Mislocalization to vacuoles may have occurred due to the overexpression of the MCA1/2(1-173) or MCA1/2 proteins, the genes of which were expressed from the strong *TDH3* promoter on a multicopy plasmid.

As described above, we detected tetramers of MCA1(1-173), MCA1(1-173)D21N, MCA2(1-173), and MCA2(1-173)D21N and explained this in the text (line 124-141, line 264-265, line 351-362, and Fig. 2). We consider this explanation to be sufficient to suggest the plasma membrane localization of these proteins.

This publication would greatly benefit from a direct prove of the oligomeric state of MCA2 when incorporate in the liposomes.

[Response] We appreciate the positive comment on our manuscript. Thank you. As mentioned above, we added Fig. 2 in order to show the oligomeric state of MCA(1-173) and explained the results in the text (line 124-141).

REVIEWERS' COMMENTS

Reviewer #2 (Remarks to the Author):

I find the revision adequate and thorough. The paper “MCAs in Arabidopsis are Ca²⁺-permeable mechanosensitive channels inherently sensitive to membrane tension” describes an important contribution to the field. The authors have carefully addressed all comments and added important data. Figures 4e and 5d now illustrate a strict rectification and complete absence of tail currents when the pipette voltage is flipped from positive to negative. This behavior of MCA2 is unique and may reflect not simply rectification of conductance but rather steep voltage dependence of the open probability. This notion, in my view, should be added to the discussion.

I have only minor suggestions:

L. 32: consider ‘gated’ instead of ‘gateable’

L. 50 and 53: please use abbreviations for two-pore K channels uniformly (TPK)

L. 60-72: It seems pertinent to mention that there is no data about potential heteromerization (co-assembly) of MCA1 and MCA2 in this paragraph.

L. 104 and 120: It might be better to use ‘protein mass’ instead of ‘protein size’

L. 169: consider ‘similar to’ instead of ‘consistent with’

In the legend to Fig. 3 please mention that the liposomes were pre-loaded with Fluo-4.

L. 209: ‘... because of strict rectification or steep voltage dependence of open probability’.

L. 219: consider ‘Reaching saturation of the open probability at higher pressures were not possible due to patch breakage’.

L. 242-245: Please break this phrase into two. ‘Therefore, activation MCA2(1-173) by tension is likely blocked independently of activation by voltage. The blockage of activation by tension appears to act through an increase of the bilayer lateral pressure imposed by these MS channel-specific blockers’

L. 248: I am not sure that LPC is an ‘inverted cone’. It is just a ‘cone’ imposing positive spontaneous curvature.

L. 259: consider ‘extracellular matrix’ instead of ‘material’

L. 269: consider ‘In the absence of tension, MCA2(1-173) is activated...’

L. 277: consider ‘natural key stimulus to open...’

Reviewer #3 (Remarks to the Author):

This version of the manuscript has address my main concerns about the proper folding and oligomerization of in vitro synthesized MCA channels.

The final publication would greatly benefit from a discussion about the rationale for the use an in vitro synthesis, vs other available purification methods.

Finally, the reason why MCA1 mechanosensitivity was not studied, should also be addressed in the discussion. Despite their sequence similarities, the differential regulation of MCA1 and MCA2 by EF hand-like and coiled-coil motifs (ref 17), as well as their distinct roles in plant physiology (ref 29), do not make this assumption so straight forward.

As stated in reference 17 "Why do these motifs appear to regulate the activity of MCA1 and MCA2 differentially, although the primary structure is 72.7% identical and 89.4% similar between the two proteins? It is possible that a slight difference in the amino acid sequence brings about a large difference in their activity." They further suggest that the heterologous expression might affect the channels differently, and that expression in mammalian cells or Arabidopsis would address this issue. Yet, a synthetic expression system was used in this study instead.

RE: manuscript NCOMMNS-20-49222 B

We would like to express our thanks to the Reviewers for their very helpful comments. We studied their comments carefully and revised the manuscript accordingly. We hope that the revised manuscript meets with their approval and is now acceptable for publication in *Nature Communications*.

Followings are point-by-point responses (**written in red**) to the Reviewers' comments.

Responses to Reviewer #2:

[This behavior of MCA2 is unique and may reflect not simply rectification of conductance but rather steep voltage dependence of the open probability. This notion, in my view, should be added to the discussion.]

We have added a sentence “This absence may also be attributed to steep voltage dependence of the open probability or activation only at positive potentials.” (Line 288-290) A similar phrase has been described in the Results section. (Line 215-216)

[L. 32: consider ‘gated’ instead of ‘gateable’]

Changed as suggested. (Line 32)

[L. 50 and 53: please use abbreviations for two-pore K channels uniformly (TPK)]

Changed as suggested. (Line 53)

[L. 60-72: It seems pertinent to mention that there is no data about potential heteromerization (co-assembly) of MCA1 and MCA2 in this paragraph.]

We have mentioned as follows: “In addition, there is no data about potential heteromerization of MCA1 and MCA2 (ref. 17). (Line 71-72)

[L. 104 and 120: It might be better to use ‘protein mass’ instead of ‘protein size’]

We have used “protein mass” instead of “protein size”. (Lines 110, 111, and 126)

[L. 169: consider ‘similar to’ instead of ‘consistent with’]

Changed as suggested. (Line 175)

[In the legend to Fig. 3 please mention that the liposomes were pre-loaded with Fluo-4.]

We have added a sentence “Liposomes were preloaded with Fluo-4.” (Line 648-649)

[L. 209: ‘... because of strict rectification or steep voltage dependence of open probability’.]

As mentioned above, we have added a sentence “This absence may also be attributed to steep voltage dependence of the open probability or activation only at positive potentials.” (Line 288-290) Similar phrase has been added in the Results section. (Line 215-216)

[L. 219: consider ‘Reaching saturation of the open probability at higher pressures were not possible due to patch breakage’.]

Changed as suggested. (Line 226-227)

[L. 242-245: Please break this phrase into two. ‘Therefore, activation MCA2(1-173) by tension is likely blocked independently of activation by voltage. The blockage of activation by tension appears to act through an increase of the bilayer lateral pressure imposed by these MS channel-specific blockers’]

The phrase has been split into two sentences. (Line 250-254)

[L. 248: I am not sure that LPC is an ‘inverted cone’. It is just a ‘cone’ imposing positive spontaneous curvature.]

Phospholipids whose head group is smaller than acyl chains are called “cone” and those whose head group is larger than acyl chains are called “inverted cone.” (see, for example, Dhara *et al*, 2020, *eLife* 9:e55152) We would like to retain the phrase “inverted cone” because the head group of LPC is larger than the acyl chains. (Line 257)

[L. 259: consider ‘extracellular matrix’ instead of ‘material’]

Changed as suggested. (Line 268)

[L. 269: consider ‘In the absence of tension, MCA2(1-173) is activated...’]

Changed as suggested. (Line 278)

[L. 277: consider 'natural key stimulus to open...']

Changed as suggested. (Line 286)

Reviewer #3 (Remarks to the Author):

[The final publication would greatly benefit from a discussion about the rationale for the use of an *in vitro* synthesis, vs other available purification methods.]

We have added a discussion about the rationale for the use of an *in vitro* synthesis in the first paragraph of the Results section. (Line 89-94)

[Finally, the reason why MCA1 mechanosensitivity was not studied, should also be addressed in the discussion. Despite their sequence similarities, the differential regulation of MCA1 and MCA2 by EF hand-like and coiled-coil motifs (ref 17), as well as their distinct roles in plant physiology (ref 29), do not make this assumption so straight forward.]

We have explained briefly the reason in the Discussion section why MCA1 mechanosensitivity was not studied. (Line 326-327).

[As stated in reference 17 "Why do these motifs appear to regulate the activity of MCA1 and MCA2 differentially, although the primary structure is 72.7% identical and 89.4% similar between the two proteins? It is possible that a slight difference in the amino acid sequence brings about a large difference in their activity." They further suggest that the heterologous expression might affect the channels differently, and that expression in mammalian cells or Arabidopsis would address this issue. Yet, a synthetic expression system was used in this study instead.]

Thank you for the comment on a possible difference in activity between MCA1 and MCA2. This comment has been reflected to explain the necessity of further studies on MCA1(1-173). (Line 323-326)

In addition, thanks to this comment, we became aware of our mistake about the percentages of amino acid identity and similarity between MCA1 and MCA2. We have corrected both in the revised manuscript. (Line 322)